# Potential Distribution of Invasive Boxwood Blight Pathogen (*Calonectria*
*pseudonaviculata*) as Predicted by Process-Based and Correlative Models

**DOI:** 10.3390/biology11060849

**Published:** 2022-06-01

**Authors:** Brittany S. Barker, Leonard Coop, Chuanxue Hong

**Affiliations:** 1Oregon Integrated Pest Management Center, Oregon State University, 4575 Research Way, Corvallis, OR 97331, USA; coopl@oregonstate.edu; 2Department of Horticulture, Oregon State University, 4017 Agriculture and Life Sciences Building, Corvallis, OR 97331, USA; 3Hampton Roads Agricultural Research and Extension Center, Virginia Polytechnic Institute and State University, 1444 Diamond Springs Road, Virginia Beach, VA 23455, USA; chhong2@vt.edu

**Keywords:** *Buxus*, invasive plant pathogen, biological invasion, climatic suitability, CLIMEX, ensemble model

## Abstract

**Simple Summary:**

Knowledge of where invasive species could potentially establish (potential distribution) is critical to prioritizing and addressing biological invasion threats. In this study, we predicted the potential distribution of *Calonectria*
*pseudonaviculata* (*Cps*), an invasive fungal pathogen that blights boxwood, an iconic landscape plant, major evergreen nursery crop, and keystone forest species. We used climate data, presence records from Europe and western Asia, and multiple modeling approaches to predict the potential distribution of *Cps* at regional and global scales and to explore the roles of temperature and moisture in shaping its distribution. Model predictions were validated using an independent presence record dataset. A consensus map of model predictions revealed that *Cps* could potentially spread and establish well beyond its currently invaded range in Europe, western Asia, New Zealand, United States and Canada. These include a number of not-yet-invaded areas in eastern and southern Europe, North America, and many regions of the world where boxwood is native. This knowledge informs policymakers and other stakeholders in these areas on the need for implementing a strict phytosanitary protocol for risk mitigation of accidental introduction, having an effective surveillance for early detection, and developing a recovery plan for the pathogen when accidental introductions occur.

**Abstract:**

Boxwood blight caused by *Cps* is an emerging disease that has had devastating impacts on *Buxus* spp. in the horticultural sector, landscapes, and native ecosystems. In this study, we produced a process-based climatic suitability model in the CLIMEX program and combined outputs of four different correlative modeling algorithms to generate an ensemble correlative model. All models were fit and validated using a presence record dataset comprised of *Cps* detections across its entire known invaded range. Evaluations of model performance provided validation of good model fit for all models. A consensus map of CLIMEX and ensemble correlative model predictions indicated that not-yet-invaded areas in eastern and southern Europe and in the southeastern, midwestern, and Pacific coast regions of North America are climatically suitable for *Cps* establishment. Most regions of the world where *Buxus* and its congeners are native are also at risk of establishment. These findings provide the first insights into *Cps* global invasion threat, suggesting that this invasive pathogen has the potential to significantly expand its range.

## 1. Introduction

Invasive plant pathogens are a global threat to the health, productivity, and diversity of plants in both agricultural and native ecosystems [1,2,3,4]. Plant pathogens including viruses, bacteria, oomycetes and fungi have been dispersing at unprecedented levels owing to increasing global trade and human travel, often remaining undetected or unidentified until they have spread and created visible impacts on hosts and recipient ecosystems [1,4,5]. In forest ecosystems, anthropogenic introductions of fungal and fungal-like pathogens are the main cause of emerging infectious diseases in trees, such as the well-known examples of chestnut blight and Dutch elm disease in North America [1,3]. Ascomycete plant pathogens that can infect multiple host species in cultivated (e.g., parks, gardens, orchards, or nurseries) and native ecosystems tend to be particularly invasive and include some of the most destructive pests of forest trees in countries with high levels of live plant trade [3,6,7].

Boxwood blight, also known as box blight, is an emerging disease of boxwood, a major evergreen shrub crop and iconic landscape plant [8,9,10], as well as a keystone forest species [11,12,13,14]. This disease is caused by two invasive ascomycete fungi, *Calonectria*
*pseudonaviculata* (*Cps*) [15] and *C. henricotiae* [16]. Both pathogen species can infect and blight boxwood foliage, resulting in rapid plant death. *Calonectria*
*henricotiae* is only known to occur in Europe, whereas *Cps* has a wider distribution that presently spans 24 countries primarily in Europe, Asia, and North America (Appendix A) [8,16,17,18]. Long-distance dispersal of *Cps* typically occurs through human-mediated transport of diseased liners (young plants) and nursery stock [8,19], often going undetected because plants can be asymptomatic until exposed to weather patterns favoring infection and subsequent symptom development [20,21]. After the initial detection of *Cps* in the United Kingdom in 1994 [22], and New Zealand in 1998 [23], the pathogen had spread to at least eight countries in continental Europe by 2013 [24]. It was first detected in western Asia in 2010, and has since become widespread throughout native *Buxus* forests in the Black Sea region of Turkey and the Caucasus [25,26,27] up to the Caspian Hyrcanian forests of northern Iran [28,29,30]. Initial reports of *Cps* in North America were from the east coast of the United States [31,32] and in Oregon [33] and British Columbia [34], and the pathogen has now been documented in at least 30 US states throughout the Southeast, Northeast, Midwest, and Pacific coast [35,36]. The geographic origin of *Cps* is unknown but is hypothesized to be in a center of diversity for *Buxus* in East Asia, the Caribbean, or Madagascar [8,18].

Boxwood blight caused by *Cps* poses a serious threat to the horticultural industry, local economies, and ecosystem integrity [8,11,20]. In the United States, the ornamental horticulture industry has sustained significant financial losses because boxwood is the number one evergreen shrub sold, with an annual wholesale value greater than USD 140 million [37]. Boxwood blight increases the cost of producing boxwood because infected plants are unsellable and must be destroyed, and controlling the disease with chemical treatments is expensive [8,36,38]. Total economic losses resulting from boxwood blight in Connecticut alone amounted to more than USD 3 million within the first year of detection [38]. Additionally, the disease has caused declines in native *Buxus* forests in western Asia, which have reduced habitat and resources for *Buxus*-associated biodiversity and negatively impacted ecosystem services such as soil stability, water quality, and flood protection [11,13,39,40]. The full host range of *Cps* is unknown; however, none of 11 tested *Buxus* species were immune to boxwood blight [41,42,43], and certain Buxaceae plants in the genera *Sarcococca* Lindl. [41,44,45] and *Pachysandra* Michx. [46,47,48] are also vulnerable to infection. Artificial inoculations demonstrated that the host range may even include plants in other taxonomic families [49]. These findings suggest that *Cps* could be a significant threat to at least some of the ca. 100 *Buxus* spp., which are primarily distributed in tropical and subtropical zones of the world, and potentially to other Buxaceae and non-Buxaceae species. Despite the rapid and ongoing spread of *Cps,* assessments of establishment risk for areas that have not (yet) been invaded are not well developed. Identifying areas that are suitable for establishment by invasive plant pathogens can guide surveillance efforts and increase the likelihood that pathogens are detected early, which is the most effective and cost-efficient method to avoid the potential ecological, economic, and societal consequences of their spread [3,6,50].

In this study, we used climatic suitability models to predict the potential distribution of *Cps* at regional and global scales and explore how climatic factors shape its known range limits. Climatic suitability models have become an important tool for assessing establishment risk for invasive fungal plant pathogens because their growth and survival is closely related to climatic conditions, particularly temperature and moisture [51,52]. Epidemics of *Cps* are often attributed to longer periods of high relative air humidity combined with warm temperatures [19,20,53,54]. We used a workflow that combined both process-based and correlative modeling algorithms to enhance the reliability of predictions and provide independent perspectives into the potential distribution of invasive species [52,55,56,57]. Our specific objectives were to identify range-limiting climatic factors for the pathogen, and to identify areas of concordance in model predictions at both regional and global scales. The models developed in this study may help with identifying locations for surveillance to detect *Cps* before it establishes and may provide insight into its potential native range.

## 2. Materials and Methods

### 2.1. Modeling Overview

First, we developed a climatic suitability model for *Cps* based on its predicted response to growth- and survival-limiting temperature and moisture factors using the “Compare Locations (1-species)” module in CLIMEX [58,59]. CLIMEX’s “Compare Locations” models (1-species and 2-species) are typically parameterized using a combination of eco-physiological data (e.g., temperature thresholds for development and survival) and point observations of occupancy or abundance from the species’ known geographical distribution [58,59]. Next, we developed climatic suitability models for *Cps* using four correlative modeling algorithms. Correlative climatic suitability models (hereafter correlative models) involve statistically linking spatial environmental data to species occurrences (presences and/or absences) to estimate the probability of other locations being part of the species distribution [57,60,61]. Compared to process-based models, correlative models require a lower number of parameters [61,62,63]. For example, they only require known distribution data as an input, whereas CLIMEX’s “Compare Locations” models require a more extensive baseline knowledge of the species. Correlative models are most reliable in predicting a species’ potential distribution in climates on which they are based and to a lesser extent for novel climates [57,64,65]. By joining process-based and correlative approaches in a combined workflow, we strived to incorporate advantages of each approach and obtain independent insights into climatic suitability for and potential distribution of *Cps*.

### 2.2. Boxwood Blight Presence Records

To fit and validate CLIMEX and correlative models, we compiled 292 presence records for *Cps* from 24 countries (Appendix A), which span the entire known distribution of the pathogen (Europe, Asia, New Zealand, and North America; Appendix A). While the use of absence records could potentially increase the robustness of models, particularly if they came from the pathogen’s well-established range in Europe, these data have not been collected to the best of our knowledge. Presence records were derived from peer-reviewed literature, theses, reports, media sources (e.g., online news articles), the Global Biodiversity Information Facility (GBIF Occurrence Download https://doi.org/10.15468/dl.44z8yr, accessed on 2 April 2021), CERIS Pest Tracker (https://pest.ceris.purdue.edu, accessed on 3 April 2021), the Agricultural Research Service Fungal Database (https://nt.ars-grin.gov/fungaldatabases, accessed on 13 October 2021), and personal communications. We excluded any record collected from garden centers or newly established plantings with boxwood plant stocks originating from another state. Ideally, positive confirmations of *Cps* should be based on both morphological and laboratory-collected data (e.g., genetic and physiological characterization) [8]; however, confirmations for a few records were based only on morphological data or the source did not provide information on the confirmation process. The likelihood that these few records were misdiagnosed is highly unlikely because boxwood blight has several diagnostic characters that distinguishes it from other boxwood diseases [8], and the records were from areas where *Cps* is known to occur.

### 2.3. CLIMEX Model

The CLIMEX model for *Cps* was generated using CLIMEX version 4.0 [58]. CLIMEX estimates the overall suitability of a location for long-term persistence by a species using the ecoclimatic index, which integrates the annual growth index (potential for population growth) with annual stresses (cold, heat, dry, and wet stress) and potentially other limiting factors that limit survival during unfavorable intervals [58,66]. Increasing ecoclimatic index values signify higher potential for growth and survival [58]. The model used a 10′ gridded global climatology centered on 1975 (1961–1990) that comes loaded with CLIMEX, which was derived from the WorldClim and CRU CL1.0 and CL2.0 datasets [67]. Eco-physiological information for *Cps* was derived from published studies on the impacts of temperature and moisture on the development and survival of the vegetative and reproductive growth stages as well as the more stress-tolerant microsclerotia stage, which can remain dormant in the top soil layers for months or even years [41,68]. As described in the next two subsections (Section 2.3.1 and Section 2.3.2), we fine-tuned CLIMEX model parameters by fitting the model to presence records from Europe and western Asia (*n* = 125), where the species may have had more time to fill its climatic niche compared to more recently invaded regions such as North America. Records from North America and New Zealand were reserved for model validation (see Section 2.6). Final CLIMEX model parameters for *Cps* are provided in Table 1.

#### 2.3.1. Temperature and Moisture Index Parameters

Four temperature index parameters in CLIMEX describe the ability for temperature-driven population growth: DV0 (limiting low temperature), DV1 (lower optimal temperature), DV2 (upper optimal temperature), and DV3 (limiting high temperature). Development of *Cps* may occur at temperatures as low as 5 °C [16,19,69], but we set DV0 to 9 °C to avoid potential biases resulting from canopy temperatures being lower than estimates from weather stations, which can produce errors in plant disease models [70]. We set DV1 and DV2 to 21 and 25 °C, respectively, because this temperature range is associated with optimal growing conditions in both field and laboratory settings [16,19,39,54,69]. We used an upper threshold of 29 °C because *Cps* colonies exhibit a low growth rate and have irregular and sclerotized morphologies at temperatures ≥ 28 °C [16,19,54].

Our unpublished re-analysis of Gehesquière (2014) [19] data indicated that 500 degree-hours during continuous leaf wetness would cause between ca. 10–50% infection for *B. sempervirens* and *B. s.* “Suffruticosa”, which is equivalent to 20 degree-days. However, CLIMEX has no way to integrate moisture with degree-day calculations; thus, we used a 10× higher value of 200 as a rough stand-in for the degree-days per generation parameter (PDD). The PDD value therefore has no true meaning with regard to actual infection conditions because it accounts only for favorable temperatures.

CLIMEX describes the overall moisture characteristic of a location using estimates of soil moisture, which combine the interactions of temperature, rainfall, and evapotranspiration. While precipitation and either high relative humidity or leaf wetness are the primary moisture drivers of *Cps* growth [20,54,71], the use of soil moisture in CLIMEX should capture the species’ response to its moisture environment in a broad sense. Four soil moisture (SM) index parameters describe the influence of moisture on population growth: SM0 (limiting low moisture), SM1 (lower optimal moisture), SM2 (upper optimal moisture), and SM3 (limiting high moisture). For each SM parameter, a value of 0 indicates no soil moisture, a value of 0.5 indicates soil moisture content is 50% of capacity, a value of 1 indicates that soil moisture content is 100% of capacity, and a value > 1 indicates a water content greater than the soil holding capacity [58]. We set SM0 to 0.2, which is higher than the permanent wilting point of plants in CLIMEX (SM0 = 0.1), because pathogens including *Cps* require free water for parts of their lifecycles. We set SM1 to 0.7 because using higher values resulted in certain presence records from more inland areas of the Black Sea and Caspian Sea regions being excluded (i.e., ecoclimatic index = 0). The upper optimal value (SM2) was set to 1.7 to ensure that wet conditions were suitable, and the upper threshold (SM3) was set to 3 to remove any constraints on growth related to high rainfall.

#### 2.3.2. Temperature and Moisture Stress Parameters

The cold and heat stress thresholds in CLIMEX (TTCS and TTHS, respectively) define the temperature below (TTCS) or above (TTHS) the stress parameter value at which stress begins to accumulate according to a weekly rate [58]. For example, if the average weekly maximum temperature (T_max_) exceeds TTHS, then heat stress = (T_max_ − TTHS) × THHS, where THHS is described by the slope of the relationship between weekly heat stress and average weekly T_max_. The threshold temperature function in CLIMEX has a multiplicative factor (referred to as “week number”) that causes stress to accumulate exponentially during consecutive weeks. To help identify appropriate TTCS and THHS values, we extracted data on minimum temperature of the coldest week (bio6) and maximum temperature of the warmest week (bio6) from the CliMond dataset for *Cps* presence records from Europe and western Asia. According to this analysis, records with the coldest temperatures, which were from northern Europe and high-elevation parts of Georgia, had weekly minimum temperatures ≥ −8 °C. This finding is consistent with temperature limits of the most cold-tolerant boxwood varieties, which are almost impossible to grow in areas where temperatures drop below −10 °C [72], and with laboratory studies of *Cps* microsclerotia survival [71,73]. We set TTCS to −10 °C and adjusted the cold stress rate (THCS) to ensure that records for *Cps* in cold areas had an ecoclimatic index that exceeded zero. Additionally, we considered maps of the northern range limit for European boxwood *B. sempervirens* (Pojark.) in Norway, which is largely confined to districts south of 62° N [74].

We set TTHS to 32 °C and adjusted the heat stress accumulation rate (THHS) such that records with the hottest temperatures, which were from northern Iran along the Caspian Sea [28,75], had ecoclimatic index values exceeding zero. Microsclerotia have been shown to survive at 40 °C for at least 24 h [73]; however, other data sources suggest that heat stress accumulates at lower temperatures. An upper lethal temperature of 33 °C has been suggested by Henricot and Culham (2002) [69] based on a laboratory study of conidial growth and by field reports from Alabama [76]. Additionally, microsclerotia died after two to five months at 30 °C under laboratory conditions [71], which if translated to field conditions, would be slightly cooler in the soil under a canopy than in weather shelters. All but a single locality record for *Cps* in Europe and western Asia occurred in areas where weekly maximum temperatures fell below 32 °C, which provides further evidence that this temperature is an appropriate heat stress threshold. 

Extremely low soil moisture reduces survival of *Cps* [54,71]. We set the dry stress threshold (SMDS) to 0.2 and weekly dry stress rate (HDS) to −0.001 because this contributed to the exclusion of the species (ecoclimatic index = 0) from relatively arid areas beyond the Black Sea and Caspian Sea regions, where boxwood does not occur [39,77]. As excessive moisture is not known to be detrimental to *Cps* survival, we used a relatively high wet stress threshold (SMWS) of 3.0 and set the rate of wet stress accumulation (HWS) to 0.005. We did not apply the hot-dry (interaction) stress parameter in CLIMEX because preliminary analyses indicated that it did not assist in modeling the potential distribution.

### 2.4. Correlative Models

#### 2.4.1. Data Inputs and Pre-Processing

Correlative modeling for *Cps* was performed using the *ENMTML* R package v. 1.0.0 [78] in R version 4.1.3 [79]. *ENMTML* provides a suite of functions to preprocess occurrence records and environmental data, fit models using a variety of algorithms, evaluate model performance for each algorithm, and combine model outputs to produce an ensemble model [78]. We fit models using presence records for Europe and western Asia, whereas records for New Zealand and North America were reserved as an independent dataset for validating predictions of presence (see Section 2.6). Prior to model fitting, we removed records for Europe and western Asia that occurred within the same grid cell using the “gridSample” function in the *dismo* R package v. 1.3.5 [80]. To reduce the potential negative effects of clustered geographic sampling on model performance [81,82], we created a subsample of 70% of records based on the expected spatial intensity function of the observed data using the “pp.subsample” function in the *spatialEco* R package v. 1.3.7 [83]. These processes resulted in 78 records (out of 125 records) for fitting correlative models (Appendix A).

Twenty-seven bioclimatic variables from the CliMond dataset [67] (https://www.climond.org, accessed on 1 June 2021) were used to generate correlative models. CliMond data were generated using the same baseline climatological inputs as CLIMEX data [67], which should increase comparability of CLIMEX and correlative model predictions. The first 19 bioclimatic variables (bio1–bio19) represent annual, weekly (interpolated from monthly), and seasonal trends and extremes in temperature and precipitation [84]. The remaining eight variables describe weekly, quarterly, and annual indices of soil moisture (bio28–bio35), which were derived by Kriticos et al. (2012) [67] using a single-bucket soil moisture model driven by the CLIMEX-formatted data. We cropped bioclimatic layers to include all of Europe and areas of western Asia extending to the eastern edge of the Hyrcanian forests in Iran (xmin = −11.5° W, xmax = 57° E, ymin = 35.6° N, ymax = 71.3° N).

A principal component analysis (PCA) based on the correlation matrix of all 27 variables was conducted because using PCA-derived variables for correlative modeling can reduce model uncertainty and increase performance of model projections into new regions [82,85,86,87]. The PCA was conducted using the “rasterPCA” function of the *RSToolbox* R package v. 0.2.6 [88] within *ENMTML*, which resulted in six principal components (PCs) that explained at least 95% of the total variance. The first and second PC axes explained the highest proportion of the total variance (49.3% + 27.3% = 76.6%) and had the strongest contributions from moisture and temperature variables, respectively (Table 2). The first PC axis (PC1) had a strong positive loading for soil moisture seasonality (bio31) and strong negative loadings for precipitation and soil moisture during warm quarters (bio18 and bio34, respectively) and for moisture during the driest week (bio30), reflecting lower warm season moisture and higher annual variation in moisture at positive PC1 scores (Table 2 and Appendix A). The second PC axis (PC2) had strong positive loadings for isothermality (bio3) and temperatures during the coldest week and quarter (bio6 and bio11, respectively), and strong negative loadings for temperature seasonality and annual range (bio4 and bio7, respectively), reflecting lower winter temperatures and higher annual variation in temperatures at positive PC2 scores. PC axes 3 through 6 explained the remaining 19.4% of total variance and were primarily related to moisture during cold and wet seasons (PC3), warm and wet season temperature (PC4), dry season precipitation and precipitation seasonality (PC5), and cold and wet season precipitation and annual precipitation (PC6).

#### 2.4.2. Model Fitting and Performance

Four machine-learning algorithms were used to fit correlative models in *ENMTML* and assess climatic variable importance. These included: boosted regression trees as implemented using the “gbm.step” function of the *dismo* R package [80,89], Gaussian process usage as implemented with the “graf” function of the *GRaF* R package v. 0.1-12 [90], Maxent as implemented with the “maxnet” function of the *maxnet* R package v. 0.1.4 [91,92], and random forest as implemented with the “randomForest” function of the *randomforest* R package v. 4.7.1 [93]. These algorithms were used because they provide powerful and efficient ways to deal with data that are nonlinear, have high dimensionality, and contain complex interactions and missing values [57,86,94]. With the possible exception of Gaussian process usage, all four methods are frequently used in species distribution modeling [52,60], and their distinctive techniques should provide unique insights into establishment risk for *Cps.* As described below, we edited the source code for *ENMTML* to change certain model parameters from default values to potentially increase model performance [95]. We used an equal number of presences and pseudo-absences (i.e., presences/absences ratio equal to 1), which were randomly allocated within a calibration area (i.e., species accessible area) delimited by a buffer of 400 km around presence records. This buffer should provide an adequate characterization of the range of climatic conditions that *Cps* could potentially experience given the distribution of native and ornamental *Buxus* spp. in Europe. 

As choices related to complexity can strongly influence correlative models [64,86,96], we used quantitative evaluations available within the *ENMeval* R package v. 2.0 [95] to determine optimal levels of complexity for the Maxent model (the package currently only implements Maxent and BIOCLIM algorithms). This included building models that used unique combinations of regularization multiplier (*rm*) values (1–5) and feature classes (linear and quadratic features vs. linear, quadratic, and hinge features), for a total of 10 models with varying levels of complexity. Models were built and cross-validated internally within *ENMeval* using the *maxnet* R package v 0.1.4 [92]. The background extent was delineated by a 400 km buffer surrounding presences. A total of 10,000 randomly sampled points were sampled from the background. Models were validated using the *k*-fold partitioning method, in which records were randomly split into *k* = 4 sets, with one set was used for model fitting and the remaining three sets used for model testing [97]. Optimal models were selected using both the cAIC and validation AUC as performance metrics [95]. According to this analysis, both metrics identified a model with linear and quadratic features as optimal, but they identified different optimal regularization multipliers (*rm* = 1 according to cAIC vs. *rm* = 2 according to AUC). We opted to use a regularization multiplier of 2 to potentially avoid model overfitting [91,96]. Thus, the Maxent model produced within *ENMTML* used a model with linear and quadratic features (“Maxent simple”) and a regularization multiplier of 2.

The boosted regression trees, random forest, and Gaussian process models used default parameters implemented from within *ENMTML* with the following exceptions. The boosted regression trees model was assigned a tree complexity (number of nodes in a tree, which control whether interactions are fitted) of 3, a bag fraction (stochasticity, which introduces randomness into the model) of 0.6, and a learning rate (determines the contribution of each tree to the growing model) of 0.0005. Bag fractions in the range of 0.5–75 have given the best results for presence–absence responses, and simple trees (tree complexity of 2 or 3) and slow learning rates (between 0.01 and 0.005) are ideal for small sample sizes [89]. The random forest model used default parameters except for the number of trees to grow and the number of variables randomly sampled as candidates at each split (*mtry*), which have a stronger influence on model accuracy than other parameters. We changed the number of trees to 1000 based on recommendations to use a large number of trees to ensure convergence and increase optimal performance [98]. The “tuneRF” function in *randomForest* was used to search for optimal *mtry* values to use for producing a forest of decision trees.

The performance for all four correlative algorithms was measured within *ENMTML*, and model outputs were combined into an ensemble model. Models were validated using the *k*-fold partitioning method and then evaluated using the area under the receiving operating characteristic curve (AUC), true skill statistics (TSS), Sørensen, and F-measure on presence-background data (F_pb_) metrics [99,100,101]. Similarity indices from community ecology including Sørensen and F_pb_ may provide better estimations of model discrimination capacity than metrics that depend on prevalence (the proportion of sites where the species is present) such as AUC [99,100,101]. An ensemble model was produced by conducting a PCA of suitability predictions (probability of occurrence) across all algorithms. Consensus models produced using PCA may outperform those produced using the weighted average method [55,102]. Some studies have shown no particular benefit to using ensemble correlative models over individual tuned models; however, they can reflect the central tendency of individual models and potentially reduce predictive uncertainty [55,102,103].

#### 2.4.3. Global Model Projections

Individual correlative models and the ensemble correlative model were projected at a global scale using the same climatic PC predictors. We avoided interpreting predictions for areas where model extrapolation into novel climates occurred because extrapolation may change the correlation structure between parameters and lead to unreliable predictions when projected outside the model calibration area [57,65,86,89]. We tested the similarity of predictors between the model calibration and global projection areas using a mobility-oriented parity (MOP) analysis [65], which is a modification and extension of the Multivariate Environmental Similarity Surface metric [57]. The MOP analysis sampled 10% of reference points from the environmental space of the calibration area and was conducted within *ENMTML* using the *kuenm* R package v. 1.1.7 [104]. Model outputs were cropped to an “environmental overlap mask” defined as areas that had MOP values ≥ 0.9. Areas where MOP values approach 0 represent strict extrapolation (complete dissimilarity of environments), whereas areas with MOP values close to 1 have highly comparable environments to those in the model calibration area [65].

### 2.5. Estimating the Potential Distribution

We overlaid predictions of the potential distribution generated by each method to produce a consensus map of the potential distribution of *Cps* at regional (Europe and western Asia, and North America (contiguous United States and bordering areas)) and global scales. In CLIMEX, any location with an ecoclimatic index greater than 0 indicates a potential for establishment [58]; however, we defined the potential distribution as areas that had an ecoclimatic index ≥ 10 because 99% of all presence records for *Cps* (i.e., those used both for model fitting and validation) met this criterion. We used a threshold suitability value of 0.3 to produce binary maps of presence/absence for each correlative model including the ensemble model. This threshold was chosen because preliminary analyses revealed that the most permissive threshold (lower presence threshold) overpredicted presence (e.g., in areas known to be too cold for the pathogen’s survival), whereas more restrictive methods appeared to underpredict presence.

### 2.6. Model Validation

The predictive accuracy of the CLIMEX model and ensemble correlative model was assessed by determining whether eight presence records from New Zealand and 159 records from North America (Appendix A) fell within the potential distribution as estimated by each method. Most records for the United States were spatially resolved only to the county or city level due to confidentiality concerns, which are typically coarser resolution than model predictions (ca. 18.5 km^2^). We therefore assessed predictions for entire counties and cities using boundary shapefiles in the *tigris* R package v. 1.6 [105].

### 2.7. Correlative Models Based on Climate Data for the Invasion Time Period

Our correlative models used historical 30-year climate normals centered on 1975 (1961–1990) because the current version of CLIMEX has no native ability to import and process other forms of gridded data, and CliMond data for more recent time frames have not been developed to the best of our knowledge. However, the invasion of *Cps* did not begin until the mid-1990s, and global temperatures and precipitation patterns have significantly changed in the past 30 years [106,107]. To explore whether climatic suitability models for *Cps* based on historical climate normals may misrepresent establishment risk, we compared correlative models produced using historical climate data (i.e., for 1961–1990) to models based on climate data for the time frame of the continental Europe invasion (2000–2020).

Climate data for both time frames were derived from the E-OBS dataset at 0.1° resolution (ca. 11.1 km^2^) [108] (https://surfobs.climate.copernicus.eu, accessed on 19 April 2022) to increase comparability of model predictions. We calculated monthly averages from daily estimates of minimum temperature, maximum temperature, and total precipitation for each time frame and then generated bioclimatic variables (bio1–19) from these data using the “biovars” function in the *dismo* R package. The E-OBS dataset lacks inputs needed to produce bioclimatic variables related to soil moisture (bio28–bio35). All models were fitted in *ENMTML* using procedures described in Section 2.4.2. However, the models used only 65 presence records owing to an absence of climate data for Iran and missing climate data for certain coastal parts of Europe.

## 3. Results

### 3.1. Correlative Model Evaluations and Variable Importance

Evaluations of the four individual correlative models and an ensemble correlative model indicated good performance (Table 3). Across the individual models, AUC ranged from 0.68 to 0.72, TSS ranged from 0.37 to 0.44, Sørensen ranged from 0.74 to 0.75, and F_pb_ ranged from 1.18 to 1.22. Evaluation metrics for the ensemble model were similar.

The PC1 variable contributed most strongly (average = 27.3%) to correlative models (Table 4), indicating an important role for warm season soil moisture and soil moisture seasonality in shaping the potential distribution of *Cps*. The PC3 variable provided the next highest contribution to correlative models (average = 23.3%), indicating that soil moisture during cold and wet seasons is another important range-limiting factor. On average, the PC2, PC4, and PC6 variables had similar contributions to models (14%, 14.2%, and 12.6%, respectively), whereas PC5 had the lowest contribution (average = 8.6%).

### 3.2. Climatic Suitability for and Potential Distribution of Cps in Europe and Western Asia

CLIMEX and correlative model predictions of climatic suitability and the potential distribution for *Cps* in Europe and western Asia were mostly concordant (Figure 1 and Figure 2b). CLIMEX predicted the highest ecoclimatic and population growth index values in the Atlantic region of western Europe, coastal areas of southern Europe, and the Black and Caspian Sea regions of western Asia (Figure 1a and Figure 3a), a finding which is consistent with predictions of high climatic suitability for these areas produced by the four correlative models (Figure 1b–e) and ensemble correlative model (Figure 1f). Consequently, these regions were included in the potential distribution according to all models (Figure 2a).

In general, all models predicted lower climatic suitability in central and eastern Europe compared to western Europe (Figure 1). However, some models predicted lower suitability in these areas than others, which resulted in discordance in predictions of the potential distribution. For example, CLIMEX included areas as far east as the western border region of Russia in the potential distribution (Figure 2b), whereas the delineation of presence according to most correlative models included few areas east of Poland (Figure 2a). Predictions of suitability and the potential distribution in Europe and western Asia among the four correlative models were largely consistent, although Maxent predicted slightly higher suitability in eastern Europe and lower suitability in northernmost regions (i.e., in Scandinavia) than other algorithms.

**Figure 1 biology-11-00849-f001:**
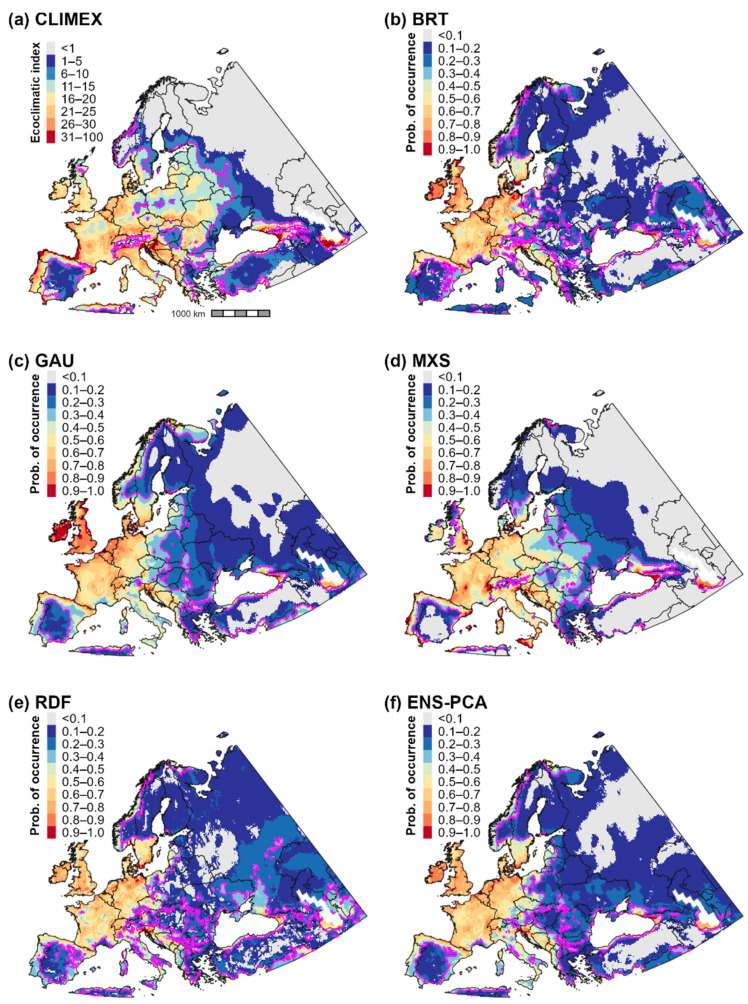
Maps of climatic suitability for *Calonectria*
*pseudonaviculata* in Europe and western Asia. Climatic suitability is estimated as the ecoclimatic index in the (**a**) CLIMEX model and as the probability of occurrence in (**b**–**f**) correlative models produced using boosted regression trees (BRT), Gaussian process (GAU), Maxent “simple” (MXS), random forest (RDF), and a principal component analysis of predictions (ensemble) produced by the four algorithms (ENS-PCA). Pink lines delineate the thresholds used to binarize models into presence–absence predictions: ecoclimatic index ≥ 10 for the CLIMEX model and probability of occurrence ≥ 0.3 for correlative models.

**Figure 2 biology-11-00849-f002:**
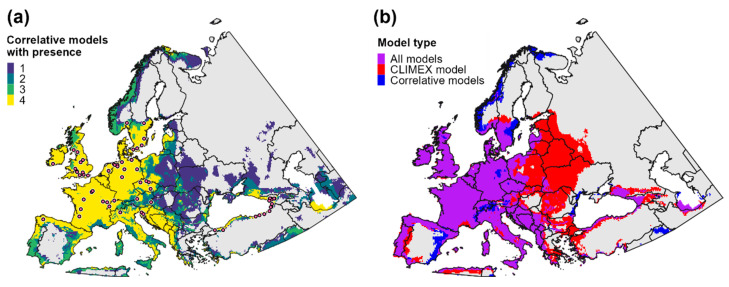
Estimates of the potential distribution for *Calonectria*
*pseudonaviculata* in Europe and western Asia. A consensus map of (**a**) presence predictions produced by individual correlative models (probability of occurrence ≥ 0.3) depicts areas of high vs. low discordance and the approximate location of presence records used for model fitting. A consensus map of (**b**) all models shows overlap in the potential distribution according to the CLIMEX model (ecoclimatic index ≥ 10) and the ensemble correlative model (purple shading) compared to areas that were included in the potential distribution by only the CLIMEX model (red shading) or by the ensemble correlative model (blue shading).

Temperature and aridity were both important range-limiting factors for *Cps* in Europe and western Asia. According to CLIMEX, cold stress is predicted to constrain *Cps* to latitudes below ca. 60° N in Europe, and it would exclude the species from western Russia except for the southernmost regions (Figure 3b). Conversely, a combination of heat stress and dry stress in Iran and countries on the eastern edge of the Caspian Sea (e.g., Turkmenistan and Kazakhstan) is predicted to limit the species to predominantly southwestern areas of the Caspian Sea region (Figure 3c,d). Heat stress and dry stress are also predicted to exclude *Cps* from most of southern Spain. 

**Figure 3 biology-11-00849-f003:**
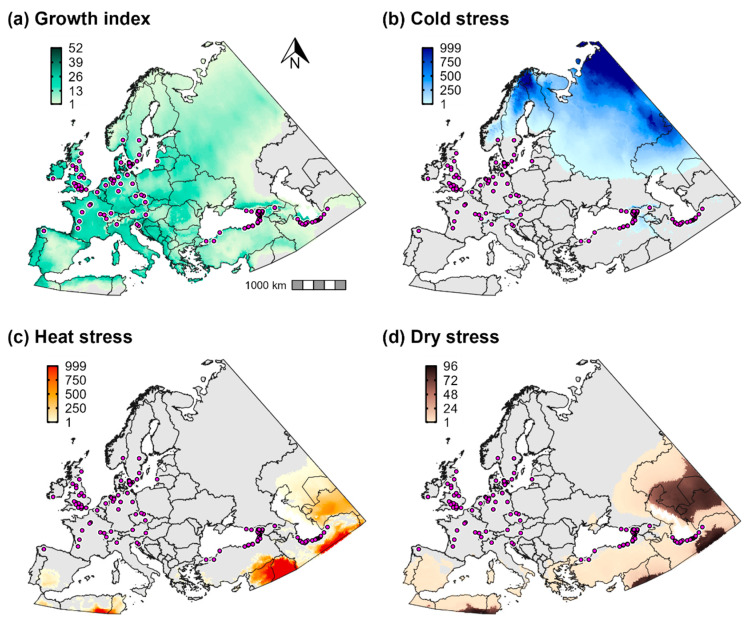
Population growth and climate stress accumulation for *Calonectria*
*pseudonaviculata* in Europe and western Asia. Population growth in CLIMEX is measured as the (**a**) annual growth index (range = 0–100). Climate stress indices (range = 0–999) include (**b**) cold stress, (**c**) heat stress, and (**d**) dry stress. Pink circles depict the approximate locations of all presence records for the region.

Our use of historical climate normals for modeling analyses does not appear to misrepresent establishment risk. Correlative models for Europe produced using E-OBS climate averages for the invasion time period (2000–2020) produced broadly concordant predictions with those developed using historical climate normals (1961–1990; Figure 4). However, predictions differed somewhat in coastal areas of southern Europe (e.g., in Spain, Italy, and Greece) and Turkey, where the correlative ensemble model for the invasion time period predicted lower suitability and a more restricted potential distribution. Similar to correlative models developed using CliMond data, models based on E-OBS data exhibited some of the highest levels of discordance in eastern Europe (Figure 4c,d).

### 3.3. Climatic Suitability for and Potential Distribution of Cps in North America

Predictions of climatic suitability and the potential distribution estimated by the CLIMEX model and correlative models were mostly concordant for eastern parts of the United States and southern Canada (Figure 5 and Figure 6). CLIMEX predicted the highest ecoclimatic and population growth index values in the eastern United States particularly in the Mid-Atlantic and Southeast regions (Figure 5a and Figure 7a), a finding which is consistent with relatively high levels of climatic suitability predicted by correlative models for these areas (Figure 5b–e). Several southeastern and midwestern states where *Cps* is not known to be established were predicted to be climatically suitable for this pathogen. These include Arkansas, Missouri, Illinois, and Indiana.

However, estimates of the potential distribution produced by the ensemble correlative model differed from the CLIMEX model at the predicted range edges in the eastern United States (Figure 6b). In the South, the ensemble correlative model included Texas and Oklahoma in the potential distribution whereas CLIMEX included only coastal areas of southeastern Texas. Additionally, the ensemble correlative model predicted a higher latitude range limit than the CLIMEX model in the Northeast and a lower latitude limit in the Midwest. Nonetheless, discordance in range limit predictions among individual correlative models indicates model uncertainty (Figure 6a). For example, only some algorithms included the entire state of Texas in the potential distribution (Gaussian process and random forest), boosted regression trees predicted a more restrictive distribution in the Northeast, and Maxent and random forest predicted a higher latitude range limit in the Midwest than other models (Figure 5b–e).

High levels of climatic suitability for *Cps* in the western United States and southern Canada was limited almost exclusively to the Pacific coast region including in southern British Columbia, western Oregon and Washington, and coastal areas of California (Figure 5). The CLIMEX model and ensemble correlative model includes these areas in the potential distribution but excluded most parts of the Intermountain West and Southwest (Figure 6b). In general, predictions of climatic suitability and presence varied to a greater extent across correlative models in the western United States (Figure 5b–e and Figure 6a). For example, Maxent predicted low suitability (probability of occurrence < 0.1) for nearly the entire region, whereas other algorithms predicted low to moderate suitability even in certain parts of the Southwest (e.g., in southern New Mexico and Arizona).

Western regions of the United States exhibited greater climatic dissimilarity to the model calibration area than eastern regions (Appendix A), with MOP values falling below 0.9 in parts of the Southwest and the Cascade Mountains in the Pacific Northwest (Figure 6b). While MOP values did not fall to levels indicative of strict model extrapolation (close to 0), they may indicate that portions of environmental space in these areas represent new combinations of predictors [65]. This finding may suggest lower predictive accuracy of correlative models for climatically dissimilar parts of the western United States.

According to CLIMEX, cold stress was the primary range-limiting factor for *Cps* in the contiguous United States and southern Canada (Figure 7b). Cold stress excluded the species from high-elevation areas in the Intermountain West (most of the Rocky Mountains), from northern parts of the Northeast (northern New York and most of Vermont, New Hampshire, and Maine) and the Midwest (most of Wisconsin and all of North Dakota, South Dakota, and Minnesota), and from southern Canada except for some coastal areas of the Pacific Ocean. However, estimates of population growth (Figure 7a) indicated that populations could grow in several of these excluded areas, a finding that suggests that *Cps* could at least temporarily establish during favorable seasons. For example, population growth was high in Wisconsin, New England, and southern parts of Ontario and Quebec; however, cold stress is predicted to prevent long-term survival throughout most of these areas.

Arid conditions in the Intermountain West and hot temperatures throughout much of the southern United States limited the pathogen’s distribution in those areas (Figure 7c,d). Whereas heat stress contributed to the exclusion of *Cps* in eastern Texas despite high population growth rates, both population growth and survival were low across most of the Southwest and Intermountain West including in western Texas, New Mexico, Colorado, Arizona, Nevada, Utah, southeastern California, and eastern Oregon and Washington.

### 3.4. Global Climatic Suitability for and Potential Distribution of Cps

Both the CLIMEX model and the ensemble correlative model predicted high climatic suitability for *Cps* in several regions of the world where the pathogen is not known to occur, such as much of Southeast Asia (e.g., in China, Japan, South Korea, Vietnam, and Indonesia), coastal areas of Australia, high elevation areas of Africa, southern parts of South America (e.g., southern Brazil, Uruguay, northern Argentina, and southern Chile) as well as the Andes region, and parts of Central America and the Caribbean (Figure 8). Additionally, all models predicted highly suitable conditions in New Zealand, where *Cps* has been reported on both the North and South Island (Appendix A). A consensus map of the global potential distribution of *Cps* indicates that establishment is possible for all of these regions (Figure 9), many of which have endemic species in the Buxaceae family (see Section 4.2). According to CLIMEX, cold temperatures were predicted to exclude the pathogen from most of northern and central Asia, whereas hot temperatures would limit establishment throughout most of northern Africa, the Middle East, India, and Australia (Appendix A).

**Figure 8 biology-11-00849-f008:**
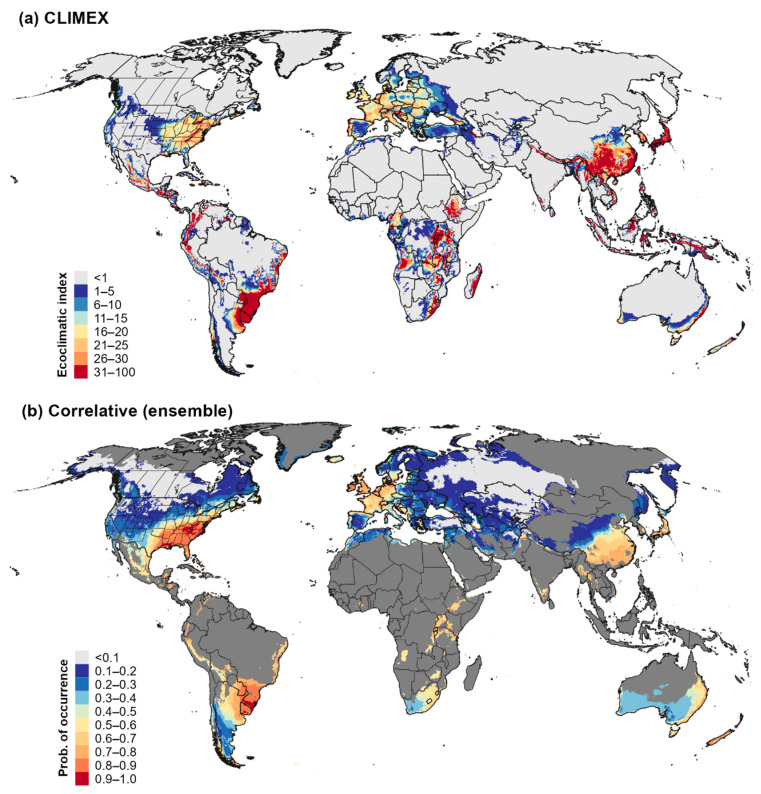
Climatic suitability for *Calonectria*
*pseudonaviculata* globally. Climatic suitability is estimated as the ecoclimatic index in the (**a**) CLIMEX model, and as the probability of occurrence in the (**b**) ensemble correlative model. Areas with relatively high levels of climatic dissimilarity to the model calibration area (MOP values < 0.9) are shown in dark gray in the correlative ensemble model.

**Figure 9 biology-11-00849-f009:**
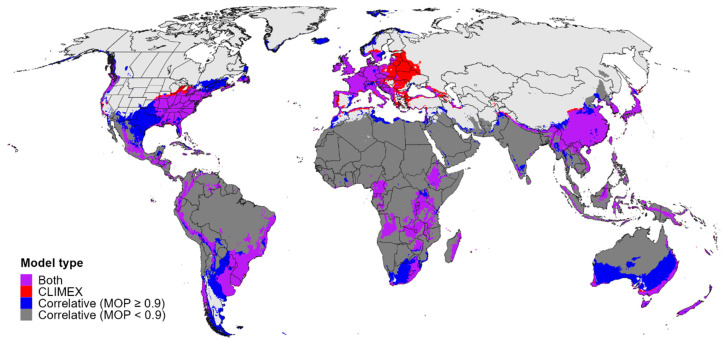
Map of the global potential distribution for *Calonectria*
*pseudonaviculata*. Areas of overlap in the potential distribution according to the CLIMEX model and the ensemble correlative model (purple shading) are shown in comparison to areas that were included in the potential distribution by only the CLIMEX model (red shading), or by the ensemble correlative model in areas with similar climates to the model calibration area (MOP values ≥ 0.9; blue shading). Correlative model predictions for areas that had dissimilar climates to the calibration area (MOP values < 0.9) and that were not included in CLIMEX’s estimate of the potential distribution (dark gray shading) were not interpreted.

### 3.5. Validation of Predictions of the Potential Distribution

Predictive accuracy of CLIMEX and ensemble correlative model estimates of the potential distribution was high. Both models included the eight New Zealand presence records in the potential distribution. All 156 records for North America were included in estimates of the potential distribution according to the CLIMEX model, whereas all but a single record (Lake County, IL, USA) were included by the ensemble correlative model.

## 4. Discussion

This study used both process-based (CLIMEX) and correlative models to assess the risk for *Cps*, a highly invasive plant pathogen, to establish at local, regional and global scales. This assessment can help guide the development of local and regional phytosanitary protocols for preventing further spread of the pathogen, prioritizing global surveillance efforts for more effective early detection, and planning for eradication, containment and management where accidental introductions occur. These three steps are critical to preventing accidental introductions of *Cps* to and becoming established in predicted high risk areas where it is not yet present [8,9]. They are also crucial to preventing boxwood blight from becoming rampant in areas where *Cps* is at its early stages of establishment [109]. The pathogen has spread rapidly, as evidenced by its invasion of 24 countries across three distant regions (Europe and western Asia, New Zealand, and North America) in less than 30 years [8,20,24]. Preventing its accidental introduction to and establishment in new areas and mitigating its local spread are both pivotal to safeguarding global boxwood crops, plantings, and forests [8,9].

All models performed well and were mostly consistent in their predictions of climatic suitability and the potential distribution for the calibration area (Europe and western Asia) and for the contiguous United States and bordering areas in North America. The CLIMEX model and ensemble correlative model correctly predicted presence for the majority of *Cps* records reserved for model validation, and evaluation metrics for the four individual correlative models and ensemble correlative model indicated good performance for the calibration area. Cold temperatures were a major range limitation at higher latitudes and elevations, as evidenced by the predicted absence of the species from northern areas that have high levels of cold stress in the CLIMEX model, and by the moderate contribution of the cold-temperature-related PC predictor (PC2) to correlative models. Moisture during warm seasons was also a major range limiting factor, as demonstrated by the absence of *Cps* from hot and arid climates in the CLIMEX model and the strong contribution of the PC predictor (PC1) related to warm season moisture and moisture seasonality to correlative models. Hot temperatures, often in combination with arid conditions, play a range-limiting role for *Cps* predominantly in the southern regions of western Asia (e.g., northern Iran) and in the southern United States.

### 4.1. Climatic Suitability for and Potential Distribution of Cps in Europe, Western Asia, and North America

Some of the highest levels of climatic suitability according to the CLIMEX model and correlative models occurred in western Europe, western Asia (Black and Caspian Sea regions), and the east coast of the United States, which is consistent with the widespread presence of *Cps* in these regions. Oceanic climates in these areas have likely facilitated the pathogen’s invasion and establishment because few gaps in precipitation and high humidity over the year combined with warm-to-hot summer temperatures creates conducive conditions for infections (Figure 10) [8,19]. In the eastern United States, *Cps* is particularly prevalent in the Mid-Atlantic and northern parts of the Southeast. There are relatively few reports of the pathogen from Florida and the Deep South (southernmost states in the Southeast), despite the inclusion of most of these regions in the potential distribution. For example, boxwood blight has not been reported beyond two locations in the Tallahassee area of northern Florida in 2016 where contaminated stock plants were received and then eradicated in 2016 [110], and to date, there have been no positive reports for Texas, Louisiana, and Mississippi [111].

According to CLIMEX, hot temperatures reduced climatic suitability throughout Florida except for along coastlines and from southern parts of the Deep South (e.g., in Alabama and Georgia), which may explain the paucity of reports from these areas. Hagan and Conner (2013) posited that disease development on container or field stock in Alabama would most likely occur during extended periods of wet weather in mid-fall into mid-spring because temperatures would be more ideal for growth than during the summer [76]. Additionally, shade can reduce temperatures and create humid conditions that may create more favorable conditions for infections in hot environments [8,112]. Additional data on the pathogen’s ability to survive prolonged heat, particularly in the more heat-resistant microsclerotia form [71,73,113], could help resolve whether it may establish in parts of the Deep South that may have ideal growing conditions during cool seasons.

Many areas with Mediterranean climates, including those in southern Europe and the Pacific coast region of the United States, were included in the potential distribution according to a consensus map of CLIMEX and ensemble correlative model predictions, but *Cps* has a limited presence in these regions to date. In southern Europe, *Cps* has been reported on *B. sempervirens* “Suffruticosa” in nurseries or gardens from only a handful of localities in northwestern Spain [114], southern France [115], northern Italy [116], and Croatia [117]. The pathogen has seemingly had opportunities to invade southern Europe given its rapid expansion throughout other parts of the continent since the early 2000s [8,20]. Host availability is unlikely an issue because boxwood is commonly grown in gardens and landscapes throughout southern Europe, and native populations of *B. sempervirens* and *B. balearica* occur in pockets in northern Africa (Morocco and Algeria), central France, the southern European peninsulas (Iberian, Italian and Balkan), certain Mediterranean Islands, and Turkey [118,119]. Models based on averages of E-OBS climate data for the invasion time period (2000–2020) predicted lower climatic suitability in parts of southern Europe than models based on historical climate normals (1961–1990), which suggests that models developed using CLIMEX and CliMond data may have overestimated establishment risk for these areas. In the western United States, *Cps* has been documented only in a handful of locations in western Oregon and the San Francisco Bay area (California) despite having a potential distribution that encompasses Mediterranean climates throughout the region, including most of the California coast and areas west of the Cascade Mountains in Oregon and Washington.

Long warm-to-hot dry summers and cool wet winters that characterize Mediterranean climates may hinder long-term establishment of *Cps* because optimal conditions for growth that transpire during warm and wet weather occur too infrequently (Figure 10). However, summer irrigation is regularly used in horticultural settings where boxwood is grown [72], and it can play a key role in *Cps* growth and survival by increasing the humidity to levels conducive for sporulation and infection [19,53,54,120]. Outbreaks in Oregon and California are often associated with summer irrigation (J. Weiland, pers. comm.) or unusually wet spring and summer weather [121]. Thus, regions with Mediterranean climates will likely be at higher risk of establishment if boxwood is irrigated during periods of optimal temperatures for *Cps* development, or during relatively wet years. Overhead irrigation in particular facilitates boxwood blight outbreaks because it creates higher relative humidity and exposes leaf surfaces to longer periods of wetness [19,120,122]. Locations that are climatically marginal for *Cps*, but which have extensive boxwood plantings, may be best able to exclude or eradicate boxwood blight outbreaks by implementing best practices such as using less dense plantings, limiting shade cover, and exclusively make use of underground irrigation [8,112,123]. Additionally, the avoidance of highly susceptible cultivars may help reduce the risk of establishment.

Climatic suitability tended to be lower in regions with humid continental climates compared to those with oceanic climates, despite the inclusion of many of these areas in the potential distribution. Optimal conditions for infections (warm and wet weather) in the humid continental climate of eastern Europe including parts of Ukraine and Russia may occur too infrequently owing to long, cold winters and warm-to-hot, dry summers. The only reports of the pathogen from these regions have come from nurseries and gardens in the Czech Republic [122,124] and a single nursery in western Ukraine [13]. The common element of diseased boxwood in gardens in the Czech Republic was the use of irrigation systems or partial-shade conditions, which created higher humidity and exposed leaves to longer periods of wetness [122]. These findings suggest that establishment risk for *Cps* in much of eastern Europe may be low in the absence of supplemental irrigation.

Climatic suitability tended to be lower in humid continental regions of North America, but cold temperatures play a larger role than aridity in reducing the risk of establishment for these areas. According to CLIMEX, cold stress lowered climatic suitability throughout interior parts of New York, New England, and southeastern Canada, which is consistent with an absence of *Cps* from these areas and with low climatic suitability predicted there by correlative models. In the midwestern United States, *Cps* has a limited presence despite the growing number of reports of the pathogen for this region, including from Missouri (2014), Kansas (2014), Illinois (2016), Indiana (2018), Arkansas (2019), Michigan (2018), and Wisconsin (2018). The establishment of the pathogen at the sites of introduction in these states is yet to be determined. If *Cps* takes hold in the Midwest, economic damages to the horticultural industry could be significant because this region is one of the top four regions in inter-regional trade of boxwood [36]. Thus, it is important for boxwood producers and users to be vigilant in watching for infections and quickly eradicating the pathogen when it is observed [50]. 

Modeling analyses indicate that cold temperatures will likely prevent establishment of *Cps* in Minnesota, North Dakota, South Dakota, and most of Wisconsin and Nebraska. The pathogen was found in North Dakota in 2019 on contaminated stock plants that were received from Ohio, but it has not been found in landscape settings where it could potentially be exposed to winter conditions (Charles Elhard, pers. comm.). Future outbreak reports from areas that are predicted to be too cold for establishment should be followed closely to assess the ability of *Cps* to overwinter. For example, soil or snow cover may offer protection to overwintering microsclerotia that may allow the pathogen to survive in areas that are predicted to be unsuitable by our models, and climate change may increase overwintering survival rates. 

Areas of Europe, western Asia, and the western United States that have arid or semi-arid climates had some of the lowest levels of climatic suitability, and will therefore be at relatively low risk of establishment at least in the absence of supplemental moisture. Range expansion of *Cps* in northern Europe and Russia will likely be prevented by cold temperatures; however, aridity often combined with hot temperatures may play a large role in limiting the pathogen’s expansion at its eastern range edge (Caspian Sea region) and southern range edge (Spain, Turkey and the Caspian Sea region). Cold temperatures were predicted to exclude *Cps* from most of Canada and the Rocky Mountains region in western North America, whereas aridity played a significant role in restricting the pathogen’s potential distribution in the Intermountain West and Southwest. Infections in these latter regions may only be possible in highly irrigated settings, and potentially in shaded areas during the hot season. With the exception of New Mexico, states in these regions have low rankings for production and total sales of boxwood [36], which could further limit the chance for *Cps* to establish there.

### 4.2. Establishment Risk for Cps in Global Centers of Diversity for Buxus and Congeners

Maps of climatic suitability and the potential distribution for *Cps* according to CLIMEX and the correlative models indicate that most regions of the world where *Buxus* and its congeners (*Didymeles*, *Haptanthus*, *Pachysandra*, *Sarcococca*, and *Styloceras*) are native are at risk of establishment. Most of the Buxaceae species are tropical or subtropical, with native ranges that include western and southern Europe, southwest, southern and eastern Asia, Africa, Madagascar, northernmost South America, Central America, Mexico and the Caribbean [125,126,127]. The consensus map of CLIMEX and ensemble correlative model predictions included much of eastern Asia and the Himalayas as *Cps*’s potential distribution areas, which are home to ca. 40 species of *Buxus* [125], four species of *Pachysandra*, and 11 species of *Sarcococca* [126]. The potential distribution in the Neotropics included the Andes region, where all five species of *Styloceras* Kunth ex A. Juss. are endemic [126], and it overlapped with at least some of the ca. 50 species of *Buxus* native to Central America and the Caribbean, such as in Mexico, Guatemala, Cuba, Hispaniola, and Puerto Rico [125,127,128]. In Africa, the potential distribution included Madagascar, which has nine endemic *Buxus* species [129], as well as other parts of Africa where *Buxus* species are native such as in South Africa, Ethiopia, Kenya, Tanzania, Angola, and Cameroon [130]. An overall lack of comprehensive and current maps that depict the ranges of Buxaceae species hinders making detailed assessments into the extent of overlap with the potential distribution of *Cps*. Nonetheless, our broad-scale assessment indicates the potential for the pathogen to expand its range globally. 

Preventing the establishment of *Cps* in regions with native boxwood is important because the pathogen can clearly cause ecological damage to affected ecosystems. Studies of *Cps* in native stands of *B. sempervirens* subsp. *colchica* in Georgia and *B. sempervirens* subsp. *hyrcana* in the Caspian Hyrcanian forests of northern Iran revealed rapid and intensive defoliation of boxwood plants of different ages, with complete defoliation occurring in up to 90% of some populations in only one year after positive detection of boxwood blight [13,28]. Infected plants are also vulnerable to attacks by secondary opportunistic pathogens that can lead to eventual death [13,14]. A literature survey showed that a loss of native boxwood in Europe and the Caucasus could lead to reductions in soil stability and subsequent declines in water quality and flood protection, changes in forest structure and composition, and declines in *Buxus*-associated biodiversity including at least 63 potentially obligate species of lichens, fungi, chromista and invertebrates [11]. Currently, there is no effective control for boxwood blight in forests because removing infected plants or applying fungicides across large areas is infeasible [13,131]. Early detection of *Cps* will therefore be the most economical and effective method to prevent additional invasions and establishments in areas with susceptible native species.

The invasion of *Cps* could be particularly devastating to species that are vulnerable both in terms of their conservation status and their susceptibility to infection. Many *Buxus* species are already threatened or endangered because of small and isolated distributions resulting from natural causes such as island endemism and post-glacial climate change [118,128], anthropogenic disturbances such as deforestation and overharvesting of wood [11], and invasions of non-native pests such as the box tree moth *Cydalima*
*perspectalis* (Walker, 1859) in Europe and western Asia [13,131,132,133]. For example, most of the *Buxus* species native to tropical America are endemic to single islands in the Caribbean [125], 37 of which occur in Cuba alone [127,128]. None of the Buxaceae species tested to date are completely immune to boxwood blight infections, although severity of disease varies widely across *Buxus* species and cultivars [41,42,43], and it appears to be low in pachysandra (*Pachysandra*) and sweet box (*Sarcococca*) species [45,134]. Susceptible species that have at least partially overlapping native ranges with the potential distribution of *Cps* include *B. sempervirens* and subspecies (southern Europe and the Black and Caspian Sea regions), *B. balearica* (Mediterranean basin), *B. bodinieri* (China), *B. glomerata* (Cuba and Hispaniola), *B. harlandii* (China to Vietnam), *B. macowanii* (South Africa), *B. riparia* (Japan), *B. wallichiana* (Himalayas from east Afghanistan to Nepal), at least three *Pachysandra* species including the endangered *P. procumbens* (eastern United States), and several *Sarcococca* species (East Asia). More studies on the susceptibility of Buxaceae species to infection are needed to better assess the risk of the pathogen establishing and causing ecological harm.

### 4.3. Potential Geographic Origin of Cps

Our global climatic suitability models for *Cps* provide some of the first insights into the potential geographic origin of the pathogen, which is still unknown [18,135]. The CLIMEX and ensemble correlative model both included a large part of southeastern China and Japan in the potential distribution, a finding that supports the hypothesis that the pathogen may have arrived to Europe on boxwood plants from East Asia [8]. A possible origin of *Cps* from China is consistent with reports that most non-European imports of *Buxus* species to Europe come from this country [136], and with a leading hypothesis for the likely origin of invasive box tree moth in Europe [137,138]. Nonetheless, we cannot rule out the possibility that *Cps* is native to another center of diversity for *Buxus* or other Buxaceae species such as in the Caribbean or Madagascar [18], particularly given that these regions were included in estimates of the potential distribution.

### 4.4. Model Uncertainty

Discordance between climatic suitability models for *Cps* in Europe, western Asia, and North America occurred predominantly at the edges of the predicted distribution, a finding consistent with observations that model uncertainty is often greater at range margins compared with range cores [56,139]. Discordance in predictions among individual correlative models can partly be explained by underlying differences in the machine-learning algorithms, each of which has unique strengths and weakness, such as different sensitivities to spatial dimensionality and correlation of environmental predictors [55,85,89,95]. For example, the boosted regression trees and random forest algorithms estimated a more restrictive potential distribution in central and eastern Europe than other algorithms, which is consistent with studies demonstrating that these methods may be prone to overfitting [60,89,90].

The threshold methods used to binarize suitability values may explain some discordance between CLIMEX and correlative model predictions of presence. For example, all models were consistent in predicting low suitability in eastern Europe; however, only CLIMEX included the majority of this region in the potential distribution. Similarly, the ensemble correlative model predicted absence in parts of southern Wisconsin with documented *Cps* detections, but all models predicted low suitability in those areas. Thus, predictions of climatic suitability by individual models should be considered when evaluating establishment risk near potential range edges.

For many regions of the world, the extrapolation of correlative models for *Cps* into areas with dissimilar climates to the model calibration area (Europe and western Asia) can explain model discordance. In particular, correlative models appeared to overpredict suitability in regions with hotter and, in some cases, wetter conditions than the calibration area. These include most equatorial regions on all continents as well as the southwestern United States. The ability for correlative models to extrapolate may decline significantly with increasing environmental distance from the calibration area, often resulting in predictions of unrealistically high levels of suitability under extreme climate values [64,65]. While the predictive performance of algorithms when extrapolating into novel climates can vary, there is likely no specific algorithm that performs well with increasing extrapolation in environmental space [64].

Process-based models such as CLIMEX are thought to be more reliable in predicting a species’ potential distribution in novel climates than correlative models because they rely on proximate constraints limiting distributions, rather than on model extrapolations [57,62,63]. For example, CLIMEX predicted that high levels of heat stress would exclude *Cps* from many hot regions where some correlative models appeared to overpredict suitability. Heat stress is measured using thresholds and rates that were calibrated using ecophysiological information and records for the pathogen in the hottest parts of its known distribution, and its predicted role in shaping the potential distribution of *Cps* seems realistic given present-day knowledge of the species.

### 4.5. Future Directions

Future climatic suitability modeling work on *Cps* could explore the potential effects of climate change on establishment risk. Models based on averages of E-OBS climate data for historical (1961–1990) and contemporary (2000–2020) time frames did not reveal marked shifts in the pathogen’s potential distribution; however, declines in climatic suitability in parts of the Mediterranean Basin are consistent with warming temperatures and increasing aridification in this region [140]. Higher minimum winter temperatures or decreased frequency or intensity of extreme cold resulting from climate change may increase rates of overwintering survival for invasive microbial pathogens [1,141], which raises the possibility that establishment risk for *Cps* at higher latitudes will increase. Additionally, increasing humidity, precipitation, and rising temperatures in certain regions such as the midwestern United States [106,107] could increase risk of establishment, whereas aridification in regions such as southern Europe, western and central Asia, and the western United States [106] may reduce risk. Climatic suitability models that account for inter-annual variations in climate may increase the accuracy of predictions under climate change because biologically relevant climatic variation that can arise from events such as droughts or heat waves may be obscured in aggregated climate datasets such as 30-year climate normals [142].

The CLIMEX model developed for this study could be modified to predict the potential distribution of *C. henricotiae*, a closely related but genetically distinct species that also causes boxwood blight [16,135]. As with *Cps*, the potential distribution and geographic origins of *C. henricotiae* are not well understood. To date, *C. henricotiae* has only been found in five countries in Europe, but further range expansion of this pathogen is expected and would likely influence boxwood blight epidemiology in the landscape because its thermotolerance is greater than *Cps* [113,135].

## 5. Conclusions

In developing species distribution models for *Cps* and evaluating the role of climatic factors in shaping its known range limits, we have provided some of the first insights into the potential invasive distribution and geographic origin of the most widespread and damaging pathogen of boxwood. Understanding where *Cps* could establish is particularly important in light of evidence for intercontinental dispersal and multiple introductions of the pathogen in the United States, which suggests that introductions are common and will likely continue to occur. While our models can assist with identifying areas to watch for *Cps* both regionally and globally, an assessment of local climates and irrigation practices for a target area may further improve insights into the likelihood of the establishment.

## Figures and Tables

**Figure 4 biology-11-00849-f004:**
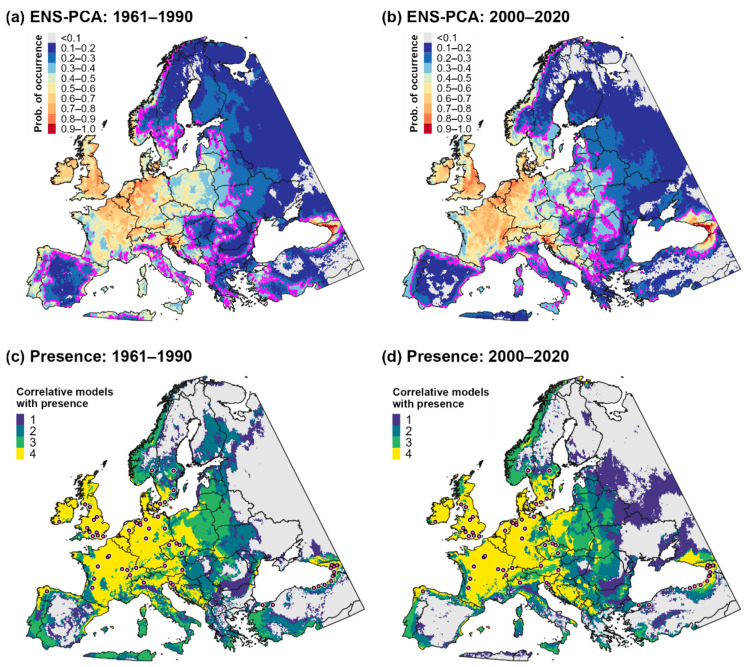
Maps of (**a**,**b**) climatic suitability and (**c**,**d**) presence for *Calonectria*
*pseudonaviculata* in Europe based on correlative models developed using E-OBS climate data for 1961–1990 and 2000–2020 (left and right panels, respectively). Pink lines in the ensemble correlative model (ENS-PCA) delineate the threshold used to binarize models into presence–absence predictions (probability of occurrence ≥ 0.3). Maps of concordance in presence predictions produced by individual correlative models show the approximate locations of records used for fitting and testing models (pink circles).

**Figure 5 biology-11-00849-f005:**
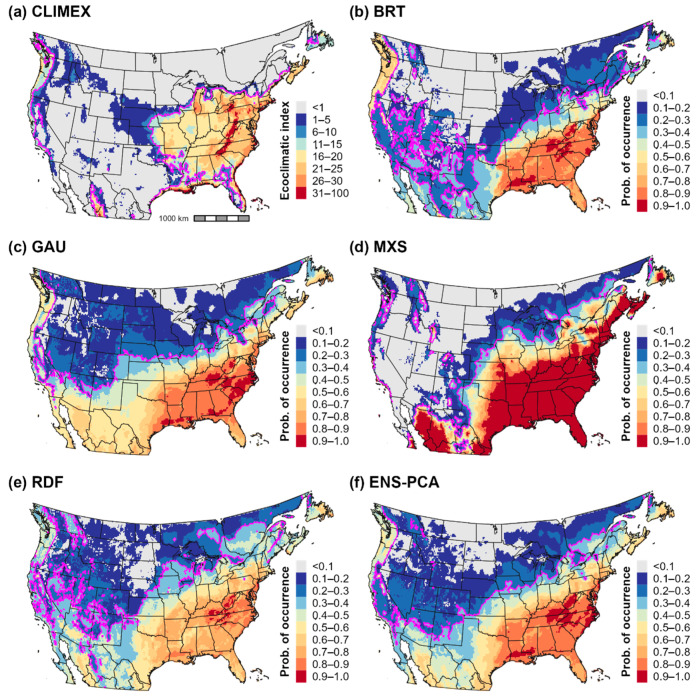
Maps of climatic suitability for *Calonectria*
*pseudonaviculata* in the contiguous United States and bordering areas in North America. Climatic suitability is estimated as the ecoclimatic index in the (**a**) CLIMEX model and as the probability of occurrence in (**b**–**f**) correlative models produced using boosted regression trees (BRT), Gaussian process (GAU), Maxent “simple” (MXS), random forest (RDF), and a principal component analysis of predictions (ensemble) produced by the four algorithms (ENS-PCA). Pink lines delineate the thresholds used to binarize models into presence–absence predictions: ecoclimatic index ≥ 10 for the CLIMEX model and probability of occurrence ≥ 0.3 for correlative models. Areas with relatively high climatic dissimilarity to the correlative model calibration area (MOP values < 0.9) are depicted in Figure 6, Figure 8 and Figure 9.

**Figure 6 biology-11-00849-f006:**
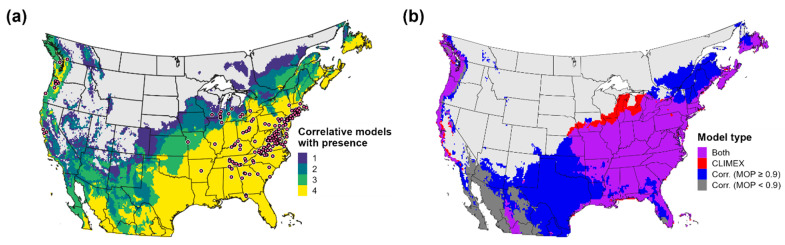
Estimates of the potential distribution for *Calonectria*
*pseudonaviculata* in the contiguous United States and bordering areas in North America. A consensus map of (**a**) presence predictions produced by individual correlative models (probability of occurrence ≥ 0.3) depicts areas of high vs. low discordance and the approximate locations of all presence records for the region (pink circles). A consensus map of (**b**) all models shows overlap in the potential distribution according to the CLIMEX model (ecoclimatic index ≥ 10) and the ensemble correlative model (purple shading) compared to areas that were included in the potential distribution by only the CLIMEX model (red shading), or by the ensemble correlative model in areas with similar climates to the model calibration area (MOP values ≥ 0.9; blue shading). Correlative model predictions for areas that had dissimilar climates to the calibration area (MOP values < 0.9) and that were not included in CLIMEX’s estimate of the potential distribution (dark gray shading) were not interpreted.

**Figure 7 biology-11-00849-f007:**
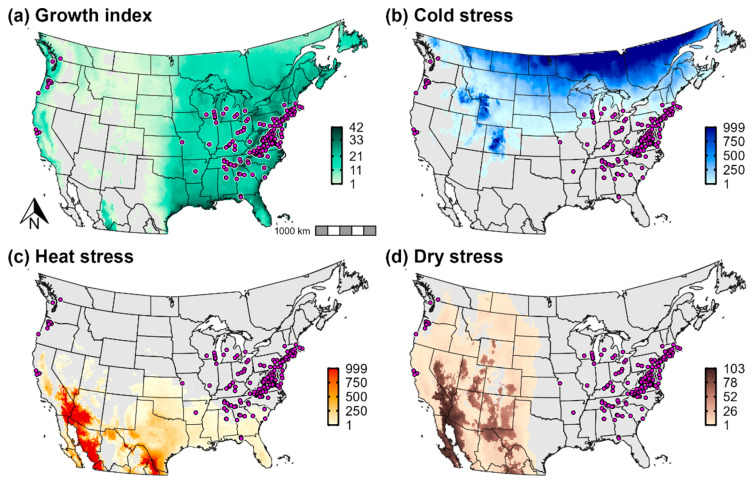
Population growth and climate stress accumulation for *Calonectria*
*pseudonaviculata* in the contiguous United States and bordering areas in North America. Population growth in CLIMEX is measured as the (**a**) annual growth index (range = 0–100). Climate stress indices (range = 0–999) include (**b**) cold stress, (**c**) heat stress, and (**d**) dry stress. Pink circles depict the approximate locations of all presence records for the region.

**Figure 10 biology-11-00849-f010:**
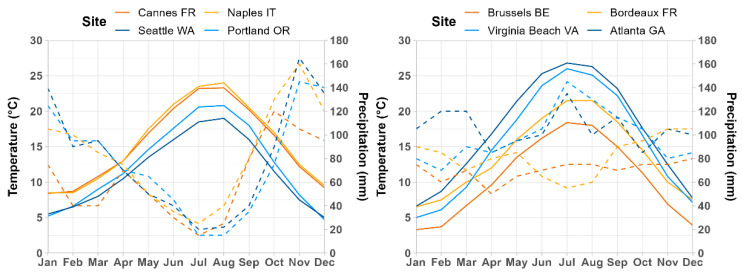
Climate comparisons for sites that are expected to differ in favorability for boxwood blight infections. Line plots depict monthly temperature (solid lines) and precipitation (dashed lines) across eight sites in Europe (orange lines) and the United States (blue lines). Sites with a Mediterranean climate (e.g., Cannes, France; Naples, Italy; Seattle, Washington; and Portland, Oregon) are less conducive for infections than sites that have higher humidity, few gaps in precipitation, and ideal temperatures for growth throughout the year, such as those in temperate/coastal climates in western Europe (e.g., Brussels, Belgium and Bordeaux, France) and warm and humid climates in the mid-Atlantic and southeastern regions of the United States (e.g., Virginia Beach, Virginia and Atlanta, Georgia). Data source: 1981–2010 climate normals, World Meteorological Organization (https://climatedata-catalogue.wmo.int; accessed on 24 September 2021).

**Table 1 biology-11-00849-t001:** CLIMEX parameter values for *Calonectria*
*pseudonaviculata*.

Parameter	Description	Value
Temperature index		
DV0	Limiting low temperature (°C)	8
DV1	Lower optimal temperature (°C)	21
DV2	Upper optimal temperature (°C)	25
DV3	Limiting high temperature (°C)	29
Moisture index		
SM0	Limiting low moisture	0.2
SM1	Lower optimal moisture	0.7
SM2	Upper optimal moisture	1.7
SM3	Limiting high moisture	3.0
Cold stress		
TTCS	Cold stress temperature threshold (°C)	−10
TCCS	Cold stress temperature rate (week^−1^)	−0.005
Heat stress		
TTHS	Heat stress temperature threshold (°C)	32
THHS	Heat stress temperature rate (week^−1^)	0.01
Dry stress		
SMDS	Dry stress threshold	0.2
HDS	Dry stress rate (week^−1^)	−0.001
Wet stress		
SMWS	Wet stress threshold	3.0
HWS	Wet stress rate (week^−1^)	0.005

**Table 2 biology-11-00849-t002:** Summary of the principal component analysis of 27 bioclimatic variables used in correlative modeling. Principal component (PC) axes were selected until the cumulative explanation proportion reached 95% or more of the total variation of the original matrix. Loadings of PCs for each variable are presented, as well as PC’s eigenvalues, the proportion of explained variance of each PC, and accumulated proportion of explained variance. The largest loadings (positive or negative) for each component (>0.30) are indicated with bold font.

Variables and Proportion of Variance	PC1	PC2	PC3	PC4	PC5	PC6
**Variable**						
Annual mean temperature (bio1)	0.10	0.26	0.06	0.29	−0.05	0.05
Mean diurnal temperature range (bio2)	0.19	−0.10	**0.31**	−0.03	−0.03	0.26
Isothermality (bio3)	0.04	**0.35**	0.11	−0.10	−0.04	0.10
Temperature seasonality (bio4)	0.11	**−0.45**	0.06	0.11	−0.07	0.05
Max temperature of warmest week (bio5)	0.22	−0.02	0.13	**0.31**	−0.09	0.12
Min temperature of coldest week (bio6)	0.01	**0.41**	−0.02	0.15	−0.02	−0.02
Temperature annual range (bio7)	0.15	**−0.44**	0.12	0.07	−0.05	0.11
Mean temperature of wettest quarter (bio8)	−0.22	−0.07	−0.15	**0.75**	0.16	−0.14
Mean temperature of driest quarter (bio9)	0.21	0.25	0.08	−0.06	−0.06	0.11
Mean temperature of warmest quarter (bio10)	0.19	0.04	0.10	**0.37**	−0.09	0.08
Mean temperature of coldest quarter (bio11)	0.03	**0.39**	0.01	0.15	−0.01	0.01
Annual precipitation (bio12)	−0.07	0.00	−0.05	0.02	−0.19	**0.33**
Precipitation of wettest week (bio13)	−0.10	−0.01	−0.05	0.06	0.06	**0.46**
Precipitation of driest week (bio14)	−0.11	0.01	0.02	0.02	**−0.45**	0.10
Precipitation seasonality (bio15)	−0.06	0.04	0.08	−0.03	**0.67**	**0.38**
Precipitation of wettest quarter (bio16)	−0.11	0.00	−0.05	0.05	0.04	**0.44**
Precipitation of driest quarter (bio17)	−0.09	0.01	0.00	0.03	**−0.45**	0.11
Precipitation of warmest quarter (bio18)	**−0.33**	−0.09	0.02	0.15	−0.09	0.15
Precipitation of coldest quarter (bio19)	0.14	0.06	−0.11	−0.07	−0.17	**0.34**
Annual mean moisture index (bio28)	−0.13	0.00	−0.26	−0.07	−0.02	0.05
Highest weekly moisture index (bio29)	0.10	−0.02	−0.47	−0.03	0.02	0.08
Lowest weekly moisture index (bio30)	**−0.34**	0.01	0.02	−0.04	−0.03	0.06
Moisture index seasonality (bio31)	**0.41**	−0.02	−0.11	0.01	0.05	0.09
Mean moisture index of wettest quarter (bio32)	0.09	−0.02	**−0.48**	−0.03	0.01	0.06
Mean moisture index of driest quarter (bio33)	**−0.33**	0.02	−0.01	−0.06	−0.04	0.04
Mean moisture index of warmest quarter (bio34)	**−0.35**	0.02	0.02	−0.06	−0.01	0.07
Mean moisture index of coldest quarter (bio35)	0.12	−0.02	**−0.51**	0.04	−0.05	0.03
**Proportion of variance**						
Proportion explained by each PC (%)	49.3	27.3	8	4.6	3.7	3.1
Accumulated proportion explained by PCs (%)	49.3	76.6	84.6	89.2	92.9	96

**Table 3 biology-11-00849-t003:** Evaluation metrics for individual correlative models and the ensemble correlative model for *Calonectria*
*pseudonaviculata*.

Algorithm	AUC	TSS	Sørensen	F_pb_
Boosted regression trees	0.68	0.37	0.74	1.18
Gaussian process	0.70	0.42	0.75	1.21
Maxent (“simple”)	0.72	0.44	0.75	1.20
Random forest	0.71	0.44	0.75	1.21
Ensemble	0.72	0.48	0.76	1.22

AUC, area under the ROC curve; TSS, true skill statistics; F_pb_, F-measure on presence-background.

**Table 4 biology-11-00849-t004:** The percent contribution of each principal component (PC) variable to correlative models produced by four algorithms. The climatic relevance of each variable (based on which bioclimatic variables had the largest loadings (positive or negative, Table 2)) and average of contributions across all algorithms is indicated.

Variable	Climatic Relevance	BRT	GAU	MXS	RDF	Avg. (%)
PC1	Warm season precipitation and soil moisture, soil moisture seasonality	26.6	38.4	24.7	19.5	27.3
PC2	Cold season temperatures, temperature seasonality	18.2	11.7	9.3	16.8	14
PC3	Cold and wet season soil moisture	23.2	24.6	27.1	18.4	23.3
PC4	Warm and wet season temperature	11.2	11.6	18.5	15.6	14.2
PC5	Dry season precipitation, precipitation seasonality	11.8	3.7	3.5	15.4	8.6
PC6	Cold and wet season precipitation, annual precipitation	9.1	10	16.9	14.3	12.6

BRT, boosted regression trees; GAU, Gaussian process; MXS, Maxent “simple”; RDF, random forest.

## Data Availability

The data, metadata, code, and derived products to reproduce the analysis and figures have been archived at Zenodo (https://doi.org/10.5281/zenodo.6567297).

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
