# Peer review of "Potential Distribution of Invasive Boxwood Blight Pathogen (Calonectriapseudonaviculata) as Predicted by Process-Based and Correlative Models"

_biology, 2022, doi:10.3390/biology11060849_

Round 1

Reviewer 1 Report

Congratulations on the great work. The methodology selected seems to be sound and is well described, as are the results. My concerns are around two main points:

  1. Should Cps be classified as invasive or as cryptogenic?
  2. The uncertainty related to the species identification for the presence records that are the base of the analysis.

Other minor comments can be found in the text.

Reviewer 2 Report

Overall, this is a good paper. I enjoyed reading it. However, it lacks to explain the potential reasons that might have caused significant different model outputs (e.g. North America). Authors used New Zealand data for validation but did not address how well the model fit for New Zealand. In addition, it does not make any recommendations or suggestions how decision makers or users should make conclusions when the model outputs are significantly different. 

Some specific comments:

  1. CLIMEX is a suite of several models. CLIMEX also includes correlation models such as Match Climates and Match Climates Regional. I believe authors used a model called "Compare Locations (1 species)". Authors need to specify the exact model used in CLIMEX. CLIMEX contains both process-based and correlative approach models. Line 121-123: For example, they only require known distribution data as input whereas CLIMEX models requires (this should be require) a more extensive baseline knowledge of the species. This is not entirely correct. CLIMEX has two models that run with only known distribution data. What you want to say is that CLIMEX models such as Compare Locations (1 species) and Compare Locations (2 species) require a more extensive baseline knowledge of the species.
  2. Authors stated that 35 bioclim variables were used in the CLIMEX model. The CLIMEX Compare Locations (1 species) model does not use bioclim variables as input data drivers. I am not sure how they would be able to use bioclim data for this model. Please specify how bioclim data are used as data drivers for the CLIMEX Compare Locations model.
  3. Line 184: It requires 500 degree-hours during the continuous leaf wetness. This is not equivalent to 20 degree-days because accumulating degree-days could happen non-continuously. I understand you used 10X higher value, but this does not solve the continuous component.
  4. Line 311: How did you estimate or generate the pseudo absence points?
  5. Line 442: What is the justification for using the probability of 0.3 as a threshold? The lower threshold value is often determined as the lowest predicted value at the 1% of the training data.
  6. Discussion: Did not much explain the potential reasons that might have caused significantly different outputs from various correlation models. This needs to be addressed in the discussion section.
  7. Soil moisture variables were two component variables that were used in the correlation models. Soil moistures were difficult to estimate. How accurate the soil moisture input data drivers are? How were these drives calculated? CLIMEX calculates soil moisture index based on the weekly precipitation and evapotranspiration.
  8. Needs some editorial reviews. 

Reviewer 3 Report

The manuscript by Dr. Barker and colleagues describes multiple species distribution modeling approaches for the fungal pathogen Calonectria pseudonaviculata, the causative agent of boxwood blight, an emerging disease that is responsible for major impacts on Buxus species around the world. Specifically, the authors use CLIMEX (a widely adopted process-based tool to infer climatic suitability for target species) and four different correlative models (boosted regression trees, Gaussian processes, Maxent, and random forest---all of which can be applied to predict the distribution of a target species based on information about its presence) to evaluate the potential distribution of the pathogen. The problem addressed in this manuscript is interesting and well suited for the readership of the journal. The analysis seems to be adequately designed and conducted, and the manuscript is very well written and organized. Except for one issue concerning the climate data used for this study, I only have some relatively minor comments that the authors may want to address while revising their work.

Major comment
As also acknowledged by the authors, one of the main limitations of this work is the use of climate data spanning a temporal window (1961--1990) that is different from that of species presence recording (1994--2021, plus some undated records). If I understand correctly, this mismatch is basically linked to the fact that CLIMEX can natively use only a specific dataset, CliMond, that does not cover more recent climate information. On the one hand, I understand the motivation of the authors to compare different species distribution modeling approaches but, on the other, I can't help but wonder whether there might exist a quantitative way to discuss the impact of the temporal mismatch that has been introduced in the analysis on the predicted species distribution. For instance, since correlative models are not bound to use CliMond as their only source, one could perhaps compare model predictions based on 1961--1990 vs. 1991--2020 climatologies using a data source that consistently covers the whole timespan. If this exercise is too time- and/or resource-consuming, it could also be limited to a single correlative model, possibly using a reduced number of covariates and a smaller spatial extent. Obviously, this is just a suggestion, but I would like to hear what the authors think about this issue.

Minor comments
- l.166: it is not completely clear to me what the authors mean by "we fine-tuned... one parameter at time by fitting..." and how this procedure relates to the description of sections 2.2.1 and 2.2.2. In general, sequential fitting of model parameters is expected to lead to inferior performances as compared to a joint calibration exercise, and the results may be influenced by the order in which parameters are calibrated. I think that this point deserves some further clarifications by the authors. 
- Figure 2: maybe I missed it, but what is the threshold used for CLIMEX, here? Is it the same as in Figure 2 (bioclimatic index > 10)?
- Figure 4: cyan lines are barely visible
- Figure 7b: I cannot understand from the map whether there exist areas with MOP values < 0.9 and predicted probability of occurrence < 0.1. If so, it might be worth making them stand out a bit. 

Other comments
- l.231: tempeartures -> temperatures
- l.234: temperatures -> temperature
